# Learning an Embedding Space for Transferable Robot Skills

**Karol Hausman**[*]
Department of Computer Science, University of Southern California
`hausman@usc.edu`

**Jost Tobias Springenberg, Ziyu Wang, Nicolas Heess, Martin Riedmiller**
DeepMind
`{springenberg,ziyu,heess,riedmiller}@google.com`

## Abstract

We present a method for reinforcement learning of closely related skills that are parameterized via a skill embedding space. We learn such skills by taking advantage of latent variables and exploiting a connection between reinforcement learning and variational inference. The main contribution of our work is an entropy-regularized policy gradient formulation for hierarchical policies, and an associated, data-efficient and robust off-policy gradient algorithm based on stochastic value gradients. We demonstrate the effectiveness of our method on several simulated robotic manipulation tasks. We find that our method allows for discovery of multiple solutions and is capable of learning the minimum number of distinct skills that are necessary to solve a given set of tasks. In addition, our results indicate that the hereby proposed technique can interpolate and/or sequence previously learned skills in order to accomplish more complex tasks, even in the presence of sparse rewards.

## 1 Introduction

Recent years have seen great progress in methods for reinforcement learning with rich function approximators, aka "deep reinforcement learning" (DRL). In the field of robotics, DRL holds the promise of automatically learning flexible behaviors end-to-end while dealing with high-dimensional, multi-modal sensor streams (Arulkumaran et al., 2017). Among these successes, there has been substantial progress in algorithms for continuous action spaces, in terms of the complexity of systems that can be controlled as well as the data-efficiency and stability of the algorithms.

Despite this recent progress, the predominant paradigm remains, however, to learn solutions from scratch for every task. Not only this is inefficient and constrains the difficulty of the tasks that can be solved, but also it limits the versatility and adaptivity of the systems that can be built. This is by no means a novel insight and there have been many attempts to address this issue (e.g. Devin et al. 2016; Rusu et al. 2016; Finn et al. 2017; Teh et al. 2017). Nevertheless, the effective discovery, representation, and reuse of skills remains an open research question.

We aim to take a step towards this goal. Our method learns manipulation skills that are continuously parameterized in an embedding space. We show how we can take advantage of these skills for rapidly solving new tasks, effectively by solving the control problem in the embedding space rather than the raw action space.

To learn skills, we take advantage of latent variables - an important tool in the probabilistic modeling literature for discovering structure in data. The main contribution of our work is an entropy-regularized policy gradient formulation for hierarchical policies, and an associated, data-efficient and robust off-policy gradient algorithm based on stochastic value gradients.

---

[*]This work was carried out during an internship at DeepMind.

Our formulation draws on a connection between reinforcement learning and variational inference and is a principled and general scheme for learning hierarchical stochastic policies. We show how stochastic latent variables can be meaningfully incorporated into policies by treating them in the same way as auxiliary variables in parametric variational approximations in inference (Salimans et al. 2014; Maaløe et al. 2016; Ranganath et al. 2016). The resulting policies can model complex correlation structure and multi-modality in action space. We represent the skill embedding via such latent variables and find that this view naturally leads to an information-theoretic regularization which ensures that the learned skills are versatile and the embedding space is well formed.

We demonstrate the effectiveness of our method on several simulated robotic manipulation tasks. We find that our method allows for the discovery of multiple solutions and is capable of learning the minimum number of distinct skills that are necessary to solve a given set of tasks. Our results indicate that the hereby proposed technique can interpolate and/or sequence previously learned skills in order to accomplish more complex tasks, even in the presence of sparse rewards. The video of our experiments is available at: `https://goo.gl/FbvPGB`.

## 2 RELATED WORK

The idea of concisely representing and re-using previously learned skills has been explored by a number of researchers, e.g. in the form of the Associative Skill Memories by Pastor et al. (2012) or the meta-level priors for generalizing the relevance of features between different manipulation skills (Kroemer & Sukhatme, 2016). Rueckert et al. (2015) use the framework of probabilistic movement primitives to extract a lower-dimensional set of primitive control variables that allow effective reuse of primitives. The use of latent variables and entropy constraints to induce diverse skills has also been considered by Daniel et al. (2016); End et al. (2017); Gabriel et al. (2017) albeit in a different framework and without using neural network function approximators. Finally Konidaris & Barto (2007) use the options framework (Sutton et al., 1999) to learn transferable options using the so-called agent-space. Inspired by these ideas, we introduce a skill embedding learning method that, by using modern DRL techniques, is able to concisely represent and reuse skills.

In the space of multi-task reinforcement learning with neural networks, Teh et al. (2017) propose a framework that allows sharing of knowledge across tasks via a task agnostic prior. Similarly, Cabi et al. (2017) make use of off-policy learning to learn about a large number of different tasks while following a main task. Denil et al. (2017) and Devin et al. (2016) propose architectures that can be reconfigured to solve a variety of tasks, and Finn et al. (2017) use meta-learning to acquire skills that can be fine-tuned effectively. Sequential learning and the need to retain previously learned skills has also been the focus of a number of researchers (e.g. Kirkpatrick et al. (2017) and Rusu et al. (2016)). In this work, we present a method that learns an explicit skill embedding space in a multi-task setting and is complementary to these works.

Our formulation draws on a connection between entropy-regularized reinforcement learning and variational inference (VI) (e.g. Todorov 2008; Toussaint 2009; Ziebart 2010; Rawlik et al. 2012; Neumann 2011; Levine & Koltun 2013; Fox et al. 2016). In particular, it considers formulations with auxiliary latent variables, a topic studied in the VI literature (e.g. Barber & Agakov 2003; Salimans et al. 2014; Ranganath et al. 2016; Maaløe et al. 2016) but not fully explored in the context of RL. The notion of latent variables in policies has been explored e.g. by controllers (Heess et al., 2016) or options (Bacon et al., 2017). Their main limitation is the lack of a principled approach to avoid a collapse of the latent distribution to a single mode. The auxiliary variable perspective introduces an information-theoretic regularizer that helps the inference model by producing more versatile behaviors. Learning versatile skills has been explored by Haarnoja et al. (2017) and Schulman et al. (2017). In particular, Haarnoja et al. (2017) learns energy-based, maximum entropy policies via the soft Q-learning algorithm. Our approach similarly uses entropy-regularized reinforcement learning and latent variables but differs in the algorithmic framework. Similar hierarchical approaches have also been studied in the work combining RL with imitation learning (Wang et al., 2017; Merel et al., 2017).

The works that are most closely related to this paper are Florensa et al. (2017); Mohamed & Rezende (2015); Gregor et al. (2016) and Hausman et al. (2017); Li et al. (2017). They use the same bound that arises in our treatment of the latent variables. Hausman et al. (2017) uses it to learn structure from demonstrations, while Mohamed & Rezende (2015); Gregor et al. (2016) use mutual informa-

tion as an intrinsic reward for option discovery. Florensa et al. (2017) follows a similar paradigm of pre-training stochastic neural network policies, which are then used to learn a new task in an on-policy setup. This approach can be viewed as a special case of the method introduced in this paper, where the skill embedding distribution is a fixed uniform distribution and an on-policy method is used to optimize the regularized objective. In contrast, our method is able to learn the skill embedding distributions, which enables interpolation between different skills as well as discovering the number of distinct skills necessary to accomplish a set of tasks. In addition, we extend our method to a more sample-efficient off-policy setup, which is important for potential applications of this method to real world environments.

## 3  Preliminaries

We perform reinforcement learning in Markov decision processes (MDP). We denote with $s \in \mathbb{R}^S$ the continuous state of the agent; $a \in \mathbb{R}^A$ denotes the action vector and $p(s_{t+1}|s_t, a_t)$ the probability of transitioning to state $s_{t+1}$ when executing action $a_t$ in $s_t$. Actions are drawn from a policy distribution $\pi_\theta(a|s)$, with parameters $\theta$; in our case a Gaussian distribution whose mean and diagonal covariance are parameterized via a neural network. At every step the agent receives a scalar reward $r(s_t, a_t)$ and we consider the problem of maximizing the sum of discounted rewards $\mathbb{E}_{\tau_\pi}[\sum_{t=0}^\infty \gamma^t r(s_t, s_t)]$.

## 4  Learning Versatile Skills

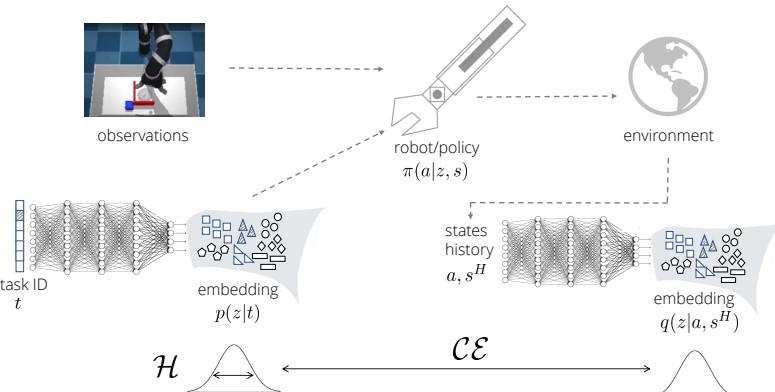

Figure 1: Schematics of our approach. We train the agent in a multi-task setup, where the task id is given as a one-hot input to the embedding network (bottom-left). The embedding network generates an embedding distribution that is sampled and concatenated with the current observation to serve as an input to the policy. After interaction with the environment, a segment of states is collected and fed into the inference network (bottom-right). The inference network is trained to classify what embedding vector the segment of states was generated from.

Before we introduce our method for learning a latent skill embedding space, it is instructive to identify the exact desiderata that we impose on the acquired skills (and thus the embedding space parameterizing them). A detailed explanation of how these goals align with recent trends in the literature is given in Section 2.

As stated in the introduction, the general goal of our method is to re-use skills learned for an initial set of tasks to speed up – or in some cases even enable – learning difficult target tasks in a transfer learning setting. We are thus interested in the following properties for the initially learned skills:

**i) generality:** We desire an embedding space, in which solutions to different, potentially orthogonal, tasks can be represented; i.e. tasks such as lifting a block or pushing it through an obstacle course should both be jointly learnable by our approach.

**ii) versatility:** We aim to learn a skill embedding space, in which different embedding vectors that are "close" to each other in the embedding space correspond to distinct solutions to the same task.

**iii) identifiability:** Given the state and action trace of an executed skill, it should be possible to identify the embedding vector that gave rise to the solution. This property would allow us to re-purpose the embedding space for solving new tasks by picking a sequence of embedding vectors.

Intuitively, the properties i)-ii) of generality and versatility can be understood as: "we hope to cover as much of the skill embedding space as possible with different clusters of task solutions, within each of which multiple solutions to the same task are represented". Property iii) intuitively helps us to: "derive a new skill by re-combining a diversified library of existing skills".

### 4.1 POLICY LEARNING VIA A VARIATIONAL BOUND ON ENTROPY REGULARIZED RL

To learn the skill-embedding we assume to have access to a set of initial tasks $\mathcal{T} = [1, \dots, T]$ with accompanying, per-task, reward functions $r_t(s, a)$, which could be comprised of different environments, variable robot dynamics, reward functions, etc.. During training time, we provide access to the task id $t \in \mathcal{T}$ (indicating which task the agent is operating in) to our RL agent. In practice – to obtain data from all training tasks for learning – we draw a task and its id randomly from the set of tasks $\mathcal{T}$ at the beginning of each episode and execute the agents current policy $\pi(a|s, t)$ in it. A conceptual diagram presenting our approach is depicted in Fig. 1.

For our policy to learn a diverse set of skills instead of just $T$ separate solutions (one per task), we endow it with a task-conditional latent variable $z$. With this latent variable, which we also refer to as "skill embedding", the policy is able to represent a distribution over skills for each task and to share these across tasks. In the simplest case, this latent variable could be resampled at every timestep and the state-task conditional policy would be defined as $\pi(a|s, t) = \int \pi(a|z, s, t)p(z|t)dz$. One simple choice would be to let $z \in 1, \dots K$, in which case the policy would correspond to a mixture of $K$ subpolicies.

Introducing a latent variable facilitates the representation of several alternative solutions but it does not mean that several alternative solutions will be learned. It is easy to see that the expected reward objective does not directly encourage such behavior. To achieve this, we formulate our objective as an entropy regularized RL problem, i.e. we maximize:

$$\max_{\pi} \mathbb{E}_{\pi, p_0, t \in \mathcal{T}} \Big[ \sum_{i=0}^{\infty} \gamma^i \big( r_t(s_i, a_i) + \alpha \mathcal{H}[\pi(a_i|s_i, t)] \big) \Big| a_i \sim \pi(\cdot|s, t), s_{i+1} \sim p(s_{i+1}|a_i, s_i) \Big], \quad (1)$$

where $p_0(s_0)$ is the initial state distribution, $\alpha$ is a weighting term – trading the arbitrarily scaled reward against the entropy – and we can define $R(a, s, t) = \mathbb{E}_{\pi}[\sum_{i=0}^{\infty} \gamma^i r_t(s_i, a_i)|s_0 = s, a_i \sim \pi(\cdot|s, t)]$ to denote the expected return for task $t$ (under policy $\pi$) when starting from state $s$ and taking action $a$. The entropy regularization term is defined as: $\mathcal{H}[\pi(a|s, t)] = \mathbb{E}_{\pi}[-\log \pi(a|s, t)]$. It is worth noting that this is very similar to the "entropy regularization" conventionally applied in many policy gradient schemes (Williams, 1992; Mnih et al., 2016) but with the critical difference that it takes into account not just the entropy of the current but also of future actions.

To apply this entropy regularization to our setting, i.e. in the presence of latent variables, extra machinery is necessary since the entropy term becomes intractable for most distributions of interest. Borrowing from the toolkit of variational inference and applying the bound from Barber & Agakov (2003), we can construct a lower bound on the entropy term from Equation (1) as (see Appendix B for details):

$$\mathbb{E}_{\pi}[-\log \pi(a|s, t)]$$
$$\geq \mathbb{E}_{\pi(a, z|s, t)} \Big[ \log \Big( \frac{q(z|a, s, t)}{\pi(a, z|s, t)} \Big) \Big] \qquad (2)$$
$$= -\mathbb{E}_{\pi_{\theta}(a|s, t)} \Big[ \mathcal{CE} \big[ p(z|a, s, t) \| q_{\psi}(z|a, s) \big] \Big] + \mathcal{H}[p_{\phi}(z|t)] + \mathbb{E}_{p_{\phi}(z|t)} \Big[ \mathcal{H}[\pi_{\theta}(a|s, z)] \Big],$$

where $q(z|a, s, t)$ is a variational inference distribution that we are free to choose, and $\mathcal{CE}$ denotes the cross entropy (CE). Note that although $p(z|a, s, t)$ is intractable, a sample based evaluation of the CE term is possible:

$$\mathbb{E}_{\pi_{\theta}(a|s, t)} \Big[ \mathcal{CE} \big[ p(z|a, s, t) \| q_{\psi}(z|a, s) \big] \Big] = \mathbb{E}_{\pi_{\theta}(a, z|s, t)} \Big[ -\log q_{\psi}(z|a, s) \Big].$$

This bound holds for any $q$. We choose $q$ such that it complies with our desired property of **identifiability** (cf. Section 4): we avoid conditioning $q$ on the task id $t$ to ensure that a given trajectory alone will allow us to identify its embedding. The above variational bound is not only valid on a single state, but can also be easily extended to a short trajectory segment of states $s_i^H = [s_{i-H}, \ldots, s_i]$, where $H$ is the segment length. We thus use the variational distribution $q_\psi(z|a, s_i^H)$ – parameterized via a neural network with parameters $\psi$. We also represent the policy $\pi_\theta(a|s, z)$ and the embedding distribution $p_\phi(z|t)$ using neural networks – with parameters $\theta$ and $\phi$ – and refer to them as policy and embedding networks respectively. The above formulation is for a single time-step; we describe a more general formulation in the Appendix (Section C).

The resulting bound meets the desiderata from Section 4: it maximizes the entropy of the embedding given the task $\mathcal{H}(p(z|t))$ and the entropy of the policy conditioned on the embedding $\mathbb{E}_{p(z|t)}\mathcal{H}(\pi(a|s, z))$ (thus, aiming to cover the embedding space with different skill clusters). The negative CE encourages different embedding vectors $z$ to have different effects in terms of executed actions and visited states: Intuitively, it will be high when we can predict $z$ from the resulting $a, s^H$. The first two terms in our bound also arise from the bound on the mutual information presented in (Florensa et al., 2017). We refer to the related work section for an in-depth discussion. We highlight that the above derivation also holds for the case where the task id is constant (or simply omitted) resulting in a bound for learning a latent embedding space encouraging the development of diverse solutions to a single task.

Inserting Equation (2) into our objective from Equation (1) yields the variational bound

$$L(\theta, \phi, \psi) = \mathbb{E}_{\substack{\pi_\theta(a, z|s, t) \\ t \in \mathcal{T}}} \left[ \sum_{i=0}^{\infty} \gamma^i \hat{r}(s_i, a_i, z, t) \middle| s_{i+1} \sim p(s_{i+1}|a_i, s_i)) \right] + \alpha_1 \mathbb{E}_{t \in \mathcal{T}} \left[ \mathcal{H}[p_\phi(z|t)] \right],$$

$$\text{where } \hat{r}(s_i, a_i, z, t) = \left[ r_t(s_i, a_i) + \alpha_2 \log q_\psi(z|a_i, s_i^H) + \alpha_3 \mathcal{H}[\pi_\theta(a|s_i, z)] \right],$$

$$(3)$$

with split entropy weighting terms $\alpha = \alpha_1 + \alpha_2 + \alpha_3$. Note that $\mathbb{E}_{t \in \mathcal{T}} \left[ \mathcal{H}[p_\phi(z|t)] \right]$ does not depend on the trajectory.

We also note that, while in the above derivation we have assumed a set of $T$ discrete tasks, our formulation does not preclude the use of a continuously parameterized tasks (e.g. parameterized via a continuous goal position instead of a task id).

## 5    LEARNING AN EMBEDDING FOR VERSATILE SKILLS IN AN OFF-POLICY SETTING

While the objective presented in Equation (3) could be optimized directly in an on-policy setting (similar to Florensa et al. (2017)), our focus in this paper is on obtaining a data-efficient, off-policy, algorithm that could, conceivably, be applied to a real robotic system in the future. The bound presented in the previous section requires environment interaction to estimate the discounted sums presented in the first three terms of Equation (3). As we will show in the following, these terms can, however, also be estimated efficiently from previously gathered data by learning a $Q$-value function[1], yielding an off-policy algorithm.

We assume the availability of a replay buffer $\mathcal{B}$ (containing full trajectory execution traces including states, actions, task id and reward), that is incrementally filled during training (see the appendix for further details). In conjunction with the trajectory traces, we also store the probabilities of each selected action and denote them with the behavior policy probability $b(a|z, s, t)$ as well as the behaviour probabilities of the embedding $b(z|t)$.

Given this replay data, we formulate the off-policy perspective of our algorithm. We start with the notion of a *lower-bound Q-function* that depends on both state $s$ and action $a$ and is conditioned on both, the embedding $z$ and the task id $t$. It encapsulates all time dependent terms from Equation (3) and can be recursively defined as:

$$Q^\pi(s_i, a_i; z, t) = \hat{r}(s_i, a_i, z, t) + \gamma \mathbb{E}_{p(s_{i+1}|a_i, s_i)}[Q^\pi(s_{i+1}, a_{i+1}; z, t)]. \qquad (4)$$

---

[1]From the perspective of variational inference, from which we are drawing inspiration in this paper, such a $Q$ function can be interpreted as an amortized inference network estimating a log-likelihood term.

To learn a parametric representation of $Q_\varphi^\pi$, we turn to the standard tools for policy evaluation from the RL literature. Specifically, we make use of the recent Retrace algorithm from Munos et al. (2016), which allows us to quickly propagate entropy augmented rewards across multiple time-steps while – at the same time – minimizing the bias that any algorithm relying on a parametric Q-function is prone to. Formally, we fit $Q_\varphi^\pi$ by minimizing the squared loss:

$$\min_\varphi \mathbb{E}_\mathcal{B}\Big[\big(Q_\varphi^\pi(s_i, a_i; z, t) - Q^{\mathrm{ret}}\big)^2\Big], \text{with}$$

$$Q^{\mathrm{ret}} = \sum_{j=i}^\infty \Big(\gamma^{j-i}\prod_{k=i}^j c_k\Big)\Big[\hat{r}(s_j, a_j, z, t) + \mathbb{E}_{\pi(a|z,s,t)}[Q_{\varphi'}^\pi(s_i, \cdot; z, t)] - Q_{\varphi'}^\pi(s_j, a_j; z, t)\Big], \quad (5)$$

$$c_k = \min\Big(1, \frac{\pi(a_k|z, s_k, t)p(z|t)}{b(a_k|z, s_k, t)b(z|t)}\Big),$$

where we compute the terms contained in $\hat{r}$ by using $r_t$ and $z$ from the replay buffer and re-compute the (cross-)entropy terms. Here, $\varphi'$ denotes the parameters of a target Q-network[2] (Mnih et al., 2015) that we occasionally copy from the current estimate $\varphi$, and $c_k$ are the per-step importance weights. Further, we bootstrap the infinite sum after $N$-steps with $E_\pi\Big[Q_{\varphi'}^\pi(s_N, \cdot; z_N, t)\Big]$ instead of introducing a $\lambda$ parameter as in the original paper (Munos et al., 2016). Equipped with this Q-function, we can update the policy and embedding network parameters without requiring additional environment interaction (using only data from the replay buffer) by optimizing the following off-policy objective:

$$\hat{L}(\theta, \phi) = \mathbb{E}_{\substack{\pi_\theta(a|z,s) \\ p_\phi(z|t) \\ s,t\in\mathcal{B}}}\left[Q_\varphi^\pi(s, a, z)\right] + \mathbb{E}_{t\in\mathcal{T}}\Big[\mathcal{H}[p_\phi(z|t)]\Big], \quad (6)$$

which can be readily obtained by inserting $Q_\varphi^\pi$ into Equation (3). To minimize this objective via gradient descent, we draw further inspiration from recent successes in variational inference and directly use the pathwise derivative of $Q_\varphi^\pi$ w.r.t. the network parameters by using the reparametrization trick (Kingma & Welling, 2013; Rezende et al., 2014). This method has previously been adapted for off-policy RL in the framework of stochastic value gradient algorithms (Heess et al., 2015) and was found to yield low-variance estimates.

For the inference network $q_\psi(z|a, s^H)$, minimizing equation (3) amounts to supervised learning, maximizing:

$$\hat{L}(\psi) = \mathbb{E}_{\substack{\pi_\theta(a,z|s,t) \\ t\in\mathcal{T}}}\left[\sum_{i=0}^\infty \gamma^i \log q_\psi(z|a, s^H)\Big| s_{i+1} \sim p_\pi(s_{i+1}|a_i, s_i)\right], \quad (7)$$

which requires sampling new trajectories to acquire target embeddings consistent with the current policy and embedding network. We found that simply re-using sampled trajectory snippets from the replay buffer works well empirically; allowing us to update all network parameters at the same time. Together with our choice for learning a Q-function, this results in a sample efficient algorithm. We refer to Section D.1 in the appendix for the derivation of the stochastic value gradient of Equation (6).

## 6 LEARNING TO CONTROL THE PREVIOUSLY-LEARNED EMBEDDING

Once the skill-embedding is learned using the described multi-task setup, we utilize it to learn a new skill. There are multiple possibilities to employ the skill-embedding in such a scenario including fine-tuning the entire policy or learning only a new mapping to the embedding space (modulating the lower level policies). In this work, we decide to focus on the latter: To adapt to a new task we freeze the policy network and only learn a new state-embedding mapping $z = f_\vartheta(x)$ via a neural network $f_\vartheta$ (parameterized by parameters $\vartheta$). In other words, we only allow the network to learn how to modulate and interpolate between the already-learned skills, but we do not allow to change the underlying policies.

---

[2]Note it will thus evaluate a different policy than the current policy $\pi$, here denoted by $b$. Nonetheless by using importance weighting via $c_k$ we are guaranteed to obtain an unbiased estimator in the limit.

## 7 EXPERIMENTAL RESULTS

Our experiments aim to answer the following questions: (1) Can our method learn versatile skills? (2) Can it determine how many distinct skills are necessary to accomplish a set of tasks? (3) Can we use the learned skill embedding for control in an unseen scenario? (4) Is it important for the skills to be versatile to use their embedding for control? (5) Is it more efficient to use the learned embedding rather than to learn to solve a task from scratch? We evaluate our approach in two domains in simulation: a point mass task to easily visualize different properties of our method and a set of challenging robot manipulation tasks. Our implementation uses 16 asynchronous workers interacting with the environment, and synchronous updates utilizing the replay buffer data.

### 7.1 DIDACTIC EXAMPLE: MULTI-GOAL POINT MASS TASK WITH SPARSE REWARDS

Similarly to Haarnoja et al. (2017), we present a set of didactic examples of multi-goal point mass tasks that demonstrate the variability of solutions that our method can discover. The first didactic example consists of a force-controlled point mass that is rewarded for being in a goal region. In order to learn the skill embedding, we use two tasks ($T = 2$), with the goals located either to the left or to the right of the initial location.

Fig. 2-bottom compares a set of trajectories produced by our method when conditioned on different Gaussian skill embedding samples with and without the variational-inference-based regularization. That is, in the latter case, we remove the cross-entropy term from the reward and train the inference network to predict embedding vectors from observed trajectories in isolation. The hereby introduced cross-entropy term between inference and embedding distributions introduces more variety to the obtained trajectories, which can be explained by the agent's incentive to help the inference network. Fig. 2-top presents the absolute error between the actual and the inferred skill embedding for both tasks. It is apparent that the trajectories generated with regularization, display more variability and are therefore easily distinguishable. This means that the inference network is able to infer the skill embedding more accurately compared to the setup without the regularization term. The constant residual error shown in the top left part of the figure corresponds to the fact that the inference network without regularization can only predict the mean of the embedding used for generating the trajectories.

The second didactic example also consists of a point mass that is rewarded for being in a goal region. However, in this experiment, we consider a case where there are four goals, that are located around the initial location (see Fig. 3 left and middle) and each of them is equally important (the agent obtains the same reward at each location) for a single task ($T = 1$). This leads to a situation where there exist multiple optimal policies for a single task. In addition, this task is challenging due to the sparsity of the rewards – as soon as one solution is discovered, it is difficult to keep exploring other goals. Due to these challenges, most existing DRL approaches would be content with finding a single solution. Furthermore, even if a standard policy gradient approach discovered multiple goals, it would have no incentive to represent multiple solutions. For this experiment, we consider both, a Gaussian embedding space as well as a multivariate Bernoulli distribution (which we expect to be more likely to capture the multi-modality of the solutions).

The left part of Fig. 3 presents the versatility of the solutions when using the multivariate Bernoulli (left) and Gaussian (middle) embedding. The multivariate Bernoulli distribution is able to discover all four solutions, whereas the Gaussian embedding focuses on discovering different trajectories that lead to only two of the goals.

In order to evaluate whether our method can determine the number of distinct skills that are necessary to accomplish a set of tasks, we conduct the following experiment. We set the number of task to four ($T = 4$) but we set two of the tasks to be exactly the same ($t = 1$ and $t = 3$). Next, we use our method to learn skill embeddings and evaluate how many distinct embeddings it learns. The results in Fig. 3-right show the KL divergence between learned embedding distributions over training iterations. One can observe that the embedding network is able to discover that task 1 and 3 can be represented by the same skill embedding resulting in the KL-divergence between these embedding distribution being close to zero ($KL(p(z|t_1)||p(z|t_3)) \approx 0)$). This indicates that the embedding network is able to discover the number of distinct skills necessary to accomplish a set of tasks.

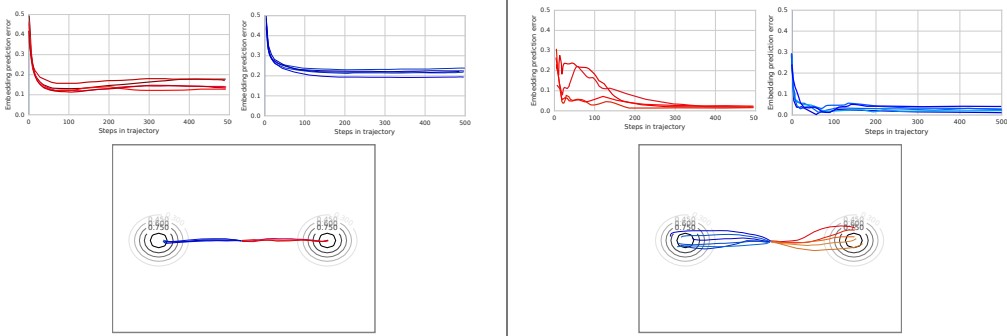

Figure 2: Bottom: resulting trajectories for different 3D embedding values with (right) and without (left) variational-inference-based regularization. The contours depict the reward gained by the agent. Top: Absolute error between the mean embedding value predicted by the inference network and the actual mean of the embedding used to generate these trajectories. Note that every error curve at the top corresponds to a single trajectory at the bottom.

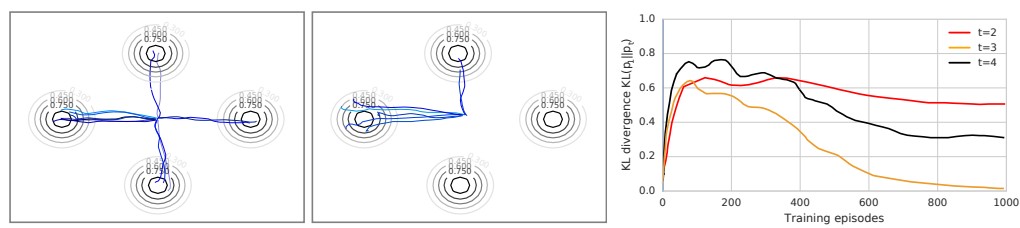

Figure 3: Left, middle: resulting trajectories that were generated by different distributions used for the skill-embedding space: multivariate Bernoulli (left), Gaussian (middle). The contours depict the reward gained by the agent. Note that there is no reward outside the goal region. Right: KL-divergence between the embedding distributions produced by task 1 and the other three tasks. Task 1 and 3 have different task ids but they are exactly the same tasks. Our method is able to discover that task 1 and 3 can be covered by the same embedding, which corresponds to the minimal KL-divergence between their embeddings.

## 7.2 CONTROL OF THE SKILL EMBEDDING FOR MANIPULATION TASKS

Next, we evaluate whether it is possible to use the learned skill embedding for control in an unseen scenario. The video of our experiments is available at: `https://goo.gl/FbvPGB`. We do so by using three simulated robotic manipulation tasks depicted in Fig. 4 and described below:

**Spring-wall.** A robotic arm is tasked to bring a block to a goal. The block is attached to a string that is attached to the ground at the initial block location. In addition, there is a short wall between the target and the initial location of the block, requiring the optimal behavior to pull the block around the wall and hold it at the goal location. The skill embedding space used for learning this skill was learned on two tasks related to this target task: bringing a block attached on a spring to a goal location (without a wall in between) and bringing a block to a goal location with a wall in between (without the spring). In order to successfully learn the new spring-wall skill, the skill-embedding space has to be able to *interpolate* between the skills it was originally trained on.

**L-wall.** The task is to bring a block to a goal that is surrounded by an L-shaped wall (see Fig. 4). The robot needs to learn how to push the block around the L-shaped wall to get to the target location. The skill embedding space used for learning this skill was learned on two tasks: push a block to a goal location (without the L-shaped wall) and lift a block to a certain height. The block was randomly spawned on a ring around the goal location that is in the center of the workspace. The purpose of this task is to demonstrate that the intuitively unrelated skills used for learning the skill embedding can be indeed beneficial for learning a new task.

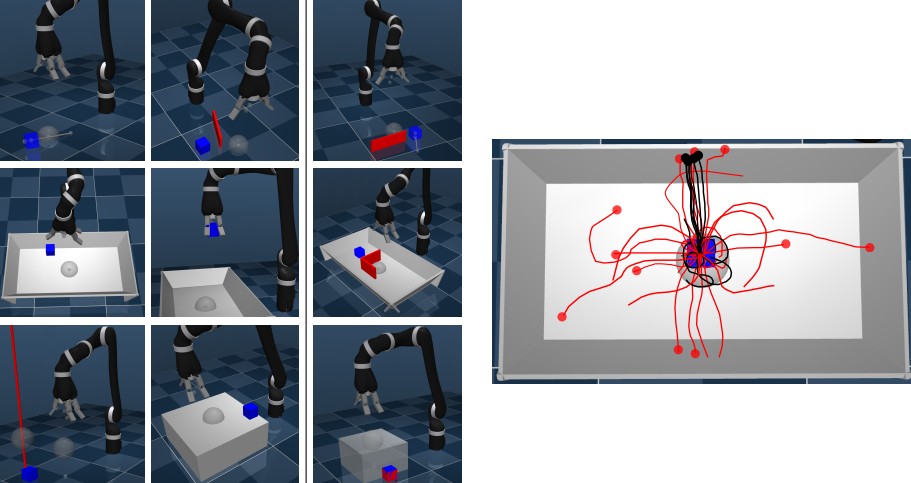

Figure 4: Left: visualization of the sequence of manipulation tasks we consider. Top row: spring-wall, middle row: L-wall, bottom row: rail-push. The left two columns depict the two initial skills that are learned jointly, the rightmost column (in the left part of the figure) depicts the transfer task that should be solved using the previously acquired skills. Right: trajectories of the block in the plane as manipulated by the robot. The trajectories are produced by sampling a random embedding vector trained with (red) and without (black) the inference network from the marginal distribution over the L-wall pre-training tasks every 50 steps and following the policy. Dots denote points at which the block was lifted.

**Rail-push.** The robot is tasked to first lift the block along the side of a white table that is firmly attached to the ground and then, to push it towards the center of the table. The initial lifting motion of the block is constrained as if the block was attached to a pole (or an upwards facing rail). This attachment is removed once the block reaches the height of the table. The skill embedding space was learned using two tasks: lift up the block attached on a rail (without the table in the scene) and push a block initialized on top of the table to its center. This task aims to demonstrate the ability to *sequence* two different skills together to accomplish a new task.

The spring-wall and L-wall tasks are performed in a setting with sparse rewards (where the only reward the robot can obtain is tied to the box being inside a small region near a target location); making them very challenging exploration problems. In contrast, the rail-push task (due to its sequential nature as well as the fact that the table acts as an obstacle) uses minor reward shaping (where we additionally reward the robot based on the distance of the box to the center of the table).

Fig. 5 shows the comparison between our method and various baselines: i) learning the transfer task from scratch, ii) learning the mapping between states and the task id ($t$) directly without a stochastic skill-embedding space, iii) learning the task by controlling the skill-embedding that was trained without variational-inference-based regularization (no inference net), iv) our method using a skill embedding obtained by pre-training on all pre-training tasks (pre-train all). A larger (6 dimensional) embedding space was used for iv).

In the spring-wall task, our approach has an advantage especially in the initial stages of training but the baseline without the inference network (no-KL in the plot) is able to achieve similar asymptotic performance. This indicates that this task does not require versatile skills and it is sufficient to find an embedding in between two skills that is able to successfully interpolate between them. It is worth noting that the remaining baselines are not able to solve this task.

For the more challenging L-wall task, our method is considerably more successful than all the baselines. This task is particularly challenging because of the set of skills that it was pre-trained on (lift the block and push the block towards the center). The agent has to discover an embedding that allows the robot to push the block along the edge of the white container - a behavior that is not directly required in any of the pre-training tasks. However, as it turns out, many successful policies for solving the lift task push the block against the wall of the container in order to perform a scooping

motion. The agent is able to discover such a skill embedding and utilize it to push the block around the L-shaped wall.

In order to investigate why the baselines are not able to find a solution to the L-wall task, we explore the embedding space produced by our method as well as by the no-inference-network baseline. In particular, we sample a random embedding vector from the marginal embedding distribution over tasks and keep it constant to generate a behavior. The resulting trajectories of the block are visualized in Fig. 4-right. One can observe that the additional regularization causes the block trajectories to be much more versatile, which makes it easier to discover a working embedding for the L-wall task.

The last task consists of a rail that the agent uses to lift the block along the wall of the table. It is worth noting that the rail task used for initial training of the rail lift skill does not include the table. For the transfer task we, however, require the agent to find a skill embedding that is able to lift the block in such a way that the arm is not in collision with the table, even though it has only encountered it in the on-table manipulation task. As shown in the most right plot of Fig. 5, such an embedding is only discovered using our method that uses variational-inference-based regularization to diversify the skills during pre-training. This indicates that due to the versatility of the learned skills, the agent is able to discover an embedding that avoids the collision with the previously unseen table and accomplishes the task successfully.

Finally, all three tasks could also be solved by our method when the embedding space is obtained by pre-training on the set of all 6 initial tasks, as indicated by the (pretrain all) curve, albeit less sample efficiently.

In summary, our algorithm is able to solve all the tasks due to having access to better exploration policies that were encoded in the skill embedding space. The consecutive images of the final policies for all three tasks are presented in Fig. 6.

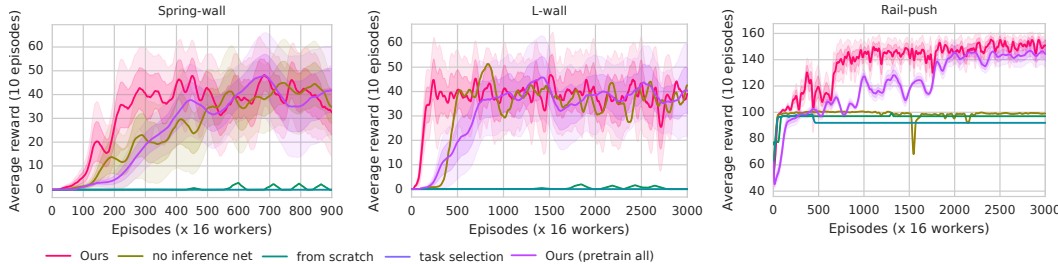

Figure 5: Comparison of our method against different training strategies for our manipulation tasks: spring-wall, L-wall, and rail-push.

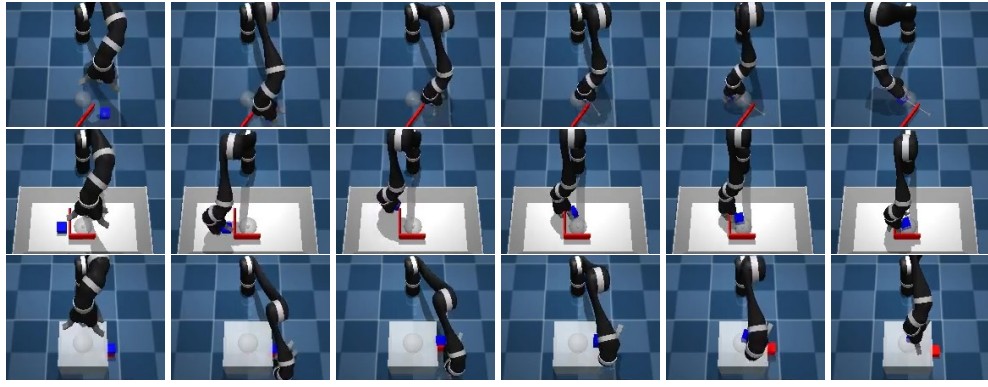

Figure 6: Final policy for all three tasks: spring-wall (top), L-wall (middle), rail-push (bottom).

## 8 CONCLUSIONS

We presented a method that learns manipulation skills that are continuously parameterized in a skill embedding space, and takes advantage of these skills by solving a new control problem in the embedding space rather than the raw action space. The skills are learned by taking advantage of latent variables and exploiting a connection between reinforcement learning and variational inference. We derived an entropy-regularized policy gradient formulation for hierarchical policies, and an associated, data-efficient off-policy algorithm based on stochastic value gradients. Our experiments indicate that our method allows for discovery of multiple solutions and is capable of learning the minimum number of distinct skills that are necessary to solve a given set of tasks. In addition, we showed that our technique can interpolate and/or sequence previously learned skills in order to accomplish more complex tasks, even in the presence of sparse rewards.

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

## A    APPENDIX

## B    VARIATIONAL BOUND DERIVATION

In order to introduce an information-theoretic regularization that encourages versatile skills, we borrow ideas from the variational inference literature. In particular, in the following, we present a lower bound of the marginal entropy $\mathcal{H}(p(x))$, which will prove useful when applied to the reinforcement learning objective in Sec. 4.1.

**Theorem 1.** *The lower bound on the marginal entropy $\mathcal{H}(p(x))$ corresponds to:*

$$\mathcal{H}(p(x)) \geq \int \int p(x,z) \log(\frac{q(z|x)}{p(x,z)}dz)dx, \tag{8}$$

*where $q(z|x)$ is the variational posterior.*

*Proof.*

$$\mathcal{H}(p(x)) = \int p(x) \log(\frac{1}{p(x)})dx = \int p(x) \log(\int q(z|x)\frac{1}{p(x)}dz)dx$$

$$= \int p(x) \log(\int q(z|x)\frac{p(z|x)}{p(x,z)}dz)dx \geq \int p(x) \int p(z|x) \log(\frac{q(z|x)}{p(x,z)}dz)dx$$

$$= \int \int p(x,z) \log(\frac{q(z|x)}{p(x,z)}dz)dx. \tag{9}$$

$\square$

## C    DERIVATION FOR MULTIPLE TIMESTEPS

We represent the trajectory as $\tau = (s_0, a_0, s_1, a_1, \ldots, s_T)$ and the learned parametrized posterior (policy) as $\pi_\theta(\tau) = p(s_0) \prod_{i=0}^{T-1} \pi_\theta(a_i|s_i)p(s_{i+1}|s_i, a_i)$. The learned inference network is represented by $q_\phi(z|\tau)$ and we introduce the pseudo likelihood that is equal to cumulative reward: $\log p(R = 1|\tau) = \sum_t r(s_t, a_t)$.

In this derivation we also assume the existence of a prior over trajectories of the form: $\mu(\tau) = p(s_0) \prod_{i=0}^{T-1} \mu(a_i|s_i)p(s_{i+1}|s_i, a_i)$. where $\mu$ represents our "prior policy". We thus consider the relative entropy between $\pi$ and $\mu$. Note that we can choose prior policy to be non-informative (e.g. a uniform prior over action for bounded action spaces).

With these definitions, we can cast RL as a variational inference problem:

$$L = \log \int p(R = 1|\tau)\mu(\tau)d\tau \geq \int \pi(\tau) \log \frac{p(R = 1|\tau)\mu(\tau)}{\pi(\tau)}d\tau$$

$$= \int \pi(\tau) \log p(R = 1|\tau)d\tau + \int \pi(\tau) \log \frac{\mu(\tau)}{\pi(\tau)}d\tau$$

$$= \mathbb{E}_\pi[\sum_t r(s_t, a_t)] + \mathbb{E}_\pi\left[\sum_t \log \frac{\mu(a_t|s_t)}{\pi(a_t|s_t)}\right]$$

$$= \mathbb{E}_\pi[\sum_t r(s_t, a_t)] + \mathbb{E}_\pi\left[\sum_t \mathrm{KL}[\pi_t||\mu_t]\right] = \mathcal{L}, \tag{10}$$

We can now introduce the latent variable $z$ that forms a Markov chain:

$$\pi(\tau) = \int \pi(\tau|z)p(z)dz$$

$$= \int p(s_0)p(z_0) \prod_{i=0}^{T-1} \pi(a_i|s_i, z_i)p(s_{i+1}|s_i, a_i)p(z_{i+1}|z_i)dz_{1:T}. \tag{11}$$

Applying it to the loss, we obtain:

$$\mathcal{L} = \mathbb{E}_\pi[\sum_t r(s_t, a_t)] + \mathbb{E}_\pi[\text{KL}[\pi(\tau)||\mu(\tau)]]$$

$$= \mathbb{E}_\pi[\sum_t r(s_t, a_t)] + \mathbb{E}_\pi\left[\log\frac{\mu(\tau)}{\int \pi(\tau|z_{1:T})p(z_{1:T})dz_{1:T}}d\tau\right]$$

$$\geq \mathbb{E}_\pi[\sum_t r(s_t, a_t)] + \mathop{\mathbb{E}}_{\pi(\tau)}\mathop{\mathbb{E}}_{p(z_{1:T}|\tau)}\left[\sum_t \int \pi(a'_t|s_t, z_t)\log\frac{\mu(a'_t|s_t)}{\pi(a'_t|s_t, z_t)}da'_t + \log\frac{q(z_{1:T}|\tau)}{p(z_{1:T})}\right].$$

$$\tag{12}$$

Equation (12) arrives at essentially the same bound as that in Equation (2) but for sequences. The exact form of (12) in the previous equation depends on the form that is chosen for $q$. For instance, for $q(z|\tau) = q(z_T|\tau)q(z_{T-1}|z_T, \tau)q(z_{T-2}|z_{T-1}, \tau)\dots$ we obtain:

$$\mathbb{E}_\pi\left[\sum_t \log\frac{\mu(a_t|s_t)}{\pi(a_t|s_t, z_t)} + \log\frac{q(z_{1:T}|\tau)}{p(z_{1:T})}\right]$$

$$=\mathbb{E}_\pi\left[\sum_t \log\frac{\mu(a_t|s_t)}{\pi(a_t|s_t, z_t)} + \sum_{t=1}^T \log\frac{q(z_{t-1}|z_t, \tau)}{p(z_{t+1}|z_t)} + \log q(z_T|\tau) - \log p(z_0)\right]. \tag{13}$$

Other forms for $q$ are also feasible, but the above form gives a nice temporal decomposition of the (augmented) reward.

## D  ALGORITHM DETAILS

### D.1  STOCHASTIC VALUE GRADIENT FOR THE POLICY

We here give a derivation of the stochastic value gradient for the objective from Equation (6) that we use for gradient based optimization. We start by reparameterizing the sampling step $z \sim p_\phi(z|t)$ for the embedding as $g_\phi(t, \epsilon_z)$, where $\epsilon_z$ is a random variable drawn from an appropriately chosen base distribution. That is, for a Gaussian embedding we can use a normal distribution (Kingma & Welling, 2013; Rezende et al., 2014) $\epsilon_z \sim \mathcal{N}(\mathbf{0}, \mathbf{I})$, where $I$ denotes the identity. For a Bernoulli embedding we can use the Concrete distribution reparametrization (Maddison et al., 2017) (also named the Gumbel-softmax trick (Jang et al., 2017)). For the policy distribution we always assume a Gaussian and can hence reparameterize using $g_\theta(t, \epsilon_a)$ with $\epsilon_a \sim \mathcal{N}(\mathbf{0}, \mathbf{I})$. Using a Gaussian embedding we then get the following gradient for the the policy parameters $\theta$

$$\nabla_\theta \hat{L}(\theta, \phi) = \nabla_\theta\left[\mathop{\mathbb{E}}_{\substack{\pi_\theta(a|z,s)\\p_\phi(z|t)\\s,t\in\mathcal{B}}}\left[Q_\varphi^\pi(s, a, z)\right] + \mathbb{E}_{t\in\mathcal{T}}\left[\mathcal{H}[p_\phi(z|t)]\right]\right],$$

$$= \mathop{\mathbb{E}}_{\substack{\epsilon_a\sim\mathcal{N}(\mathbf{0},\mathbf{I})\\\epsilon_z\sim\mathcal{N}(\mathbf{0},\mathbf{I})\\s,t\in\mathcal{B}}}\left[\nabla_\theta\, Q_\varphi^\pi(s, g_\theta(t, \epsilon_a), g_\phi(t, \epsilon_z))\nabla_\theta g_\theta(t, \epsilon_a)\right],$$

$$\tag{14}$$

and, for the embedding network parameters,

$$\nabla_\phi \hat{L}(\theta, \phi) = \nabla_\phi\left[\mathop{\mathbb{E}}_{\substack{\pi_\theta(a|z,s)\\p_\phi(z|t)\\s,t\in\mathcal{B}}}\left[Q_\varphi^\pi(s, a, z)\right] + \mathbb{E}_{t\in\mathcal{T}}\left[\mathcal{H}[p_\phi(z|t)]\right]\right],$$

$$= \mathop{\mathbb{E}}_{\substack{\epsilon_a\sim\mathcal{N}(\mathbf{0},\mathbf{I})\\\epsilon_z\sim\mathcal{N}(\mathbf{0},\mathbf{I})\\s,t\in\mathcal{B}}}\left[\nabla_g\, Q_\varphi^\pi(s, g_\theta(t, \epsilon_a), g_\phi(t, \epsilon_z))\nabla_\phi g_\phi(t, \epsilon_z)\right] + \mathbb{E}_{t\in\mathcal{T}}\left[\nabla_\phi\mathcal{H}[p_\phi(z|t)]\right].$$

$$\tag{15}$$

# E  IMPLEMENTATION DETAILS

## E.1  TASK STRUCTURE

All the tasks presented in Sec. 7.2 share a similar structure, in that the observation space used for the pre-trained skills and the observation space used for the final task are the same. For all three tasks, the observations include: joint angles (7) and velocities (7) of the robot joints, the position (3), orientation (4) and linear velocity (3) of the block as well as the position of the goal (3). The action space is also the same across all tasks and consists of joint torques for all the robot joints (7). We choose such a structure (making sure that the action space matches and providing only proprioceptive information to the policy) to make sure we i) can transfer the policy between tasks directly; 2) to ensure that the only way the agent is informed about changing environment dynamics (e.g., the attachment of the block to a string, the existence of a wall, etc.) is through the task id.

The rationale behind having the same observation space between the pre-trained skills and the final task comes from the fact that currently, our architecture expects the same observations for the final policy over embeddings and the skill subpolicies. We plan to address this limitation in future work.

## E.2  NETWORK ARCHITECTURE AND HYPERPARAMETERS

The hereby presented values were used to generate results for the final three manipulation tasks presented in Sec. 7.2. For both policy and inference network we used two-layer fully connected neural networks with exponentiaded linear activations (Clevert et al., 2015) (for layer sizes see table) to parameterize the distribution parameters. As distributions we always relied on a gaussian distribution $\mathcal{N}(\mu_\theta(x), \mathrm{diag}(\sigma(x)))$ whose mean and diagonal covariance are parameterized by the policy network via $[mu(x), \log \sigma(x)] = f_\theta(x)$. For the embedding network the mapping from one-hot task vectors to distribution parameters is given via a linear transformation. For the inference network we map to the parameters of the same distribution class via another neural network.

| Hyperparameter | Spring-wall | L-wall | Rail-push |
|---|---|---|---|
| State dims | 27 | 27 | 27 |
| Action dims | 7 | 7 | 7 |
| Policy net | 200-100 | 200-100 | 200-100 |
| Q function net | 200-200 | 200-200 | 200-100 |
| Inference net | 200-200 | 200-200 | 200-100 |
| Embedding distribution | 3D Gaussian | 3D Gaussian | 3D Gaussian |
| Minibatch size (per-worker) | 32 | 32 | 32 |
| Replay buffer size | $1e^5$ | $1e^5$ | $1e^5$ |
| $\alpha_1$ | $1e^3$ | $1e^3$ | $1e^3$ |
| $\alpha_2$ | $1e^3$ | $1e^3$ | $1e^3$ |
| $\alpha_3$ | $1e^3$ | $1e^3$ | $1e^3$ |
| Discount factor ($\gamma$) | 0.99 | 0.99 | 0.99 |
| Adam learning rate | $1e^{-3}$ | $1e^{-3}$ | $1e^{-3}$ |

