# OpenReview forum: "Learning an Embedding Space for Transferable Robot Skills"
_ICLR.cc/2018/Conference — Accept (Poster)_

### Official Review · AnonReviewer3 · 2017-11-26
**I find the method to be theoretically interesting and valuable to the learning community. However, the experiments are not entirely convincing.**

**Rating:** 7
**Confidence:** 4

**Review:**

In this paper, (previous states, action) pairs and task ids are embedded into the same latent space with the goal of generalizing and sharing across skill variations. Once the embedding space is learned, policies can be modified by passing in sampled or learned embeddings.

Novelty and Significance: To my knowledge, using a variational approach to embedding robot skills is novel. Significantly, the embedding is learned from off-policy trajectories, indicating feasibility on a real-world setting. The manipulation experiments show nice results on non-trivial tasks. However, no comparisons are shown against prior work in multitask or transfer learning. Additionally, the tasks used to train the embedding space were tailored exactly to the target task, making it unclear that this method will work generally.

Questions:
- I am not sure how to interpret Figure 3. Do you use Bernoulli in the experiments?
- How many task IDs are used for each experiment? 2?
- Are the manipulation experiments learned with the off-policy variant?
- Figure 4b needs the colors to be labeled. Video clips of the samples would be a plus.
- (Major) For the experiments, only exactly the useful set of tasks is used to train the embedding. What happens if a single latent space is learned from all the tasks, and Spring-wall, L-wall, and Rail-push are each learned from the same embedding.

I find the method to be theoretically interesting and valuable to the learning community. However, the experiments are not entirely convincing.

---

> ### Author Response · Authors · 2018-01-05
> **Response to Reviewer 3**
>
> We are grateful for the insightful comments and suggestions.
>
> Please find the answers to the inline questions below, we will clarify all of these points in the final version of the paper.
> - I am not sure how to interpret Figure 3. Do you use Bernoulli in the experiments?
> - A Bernoulli distribution is only used for for Figure 3 to demonstrate that our method can work with other distributions.
>
> - How many task IDs are used for each experiment? 2?
> - Yes, T was set to 2 for the manipulation experiments.
>
> - Are the manipulation experiments learned with the off-policy variant?
> - That is correct. All experiments were performed in an off-policy setting. This decision was made due to the higher sample-efficiency of the off-policy methods.
>
> - Figure 4b needs the colors to be labeled. Video clips of the samples would be a plus.
> - We will add the labels and address this problem in the final version of the paper
>
> Regarding the last question on training the embedding space on all of the tasks; we are currently working on this experiment and are planning to include it in the final version of the paper. It is worth noting that the multi-task RL training can be challenging (especially with poorly scaled rewards) and it maintains as an open problem that is beyond the scope of this work. Our method presents a solution to a problem of finding an embedding space that enables re-using, interpolating and sequencing previously learned skills, with the assumption that the RL agent was able to learn them in the first place. However, we strongly believe that the off-policy setup presented in this work has much more flexibility that its on-policy equivalents as to how to address the multi-task RL problem.

---

> ### Comment · AnonReviewer3 · 2018-01-13
> **Revision of Review**
>
> I would really like to see an experiment where an embedding space is trained on a wider variety of tasks rather than just what is needed to generalize to the target task. However, I find that this paper is a valuable contribution to ICLR, and I think that it should be accepted.
>
> As ICLR allows the authors to upload a new pdf, I do not understand why the author response only said they would make changes in the final version (especially for things like labeling a figure).

---

> > ### Author Response · Authors · 2018-01-17
> > **Manuscript updated**
> >
> > We would you like to notify the reviewer that the pdf has been updated with the requested changes including the new experiment with the embedding pre-trained on all 6 tasks.

---

### Official Review · AnonReviewer2 · 2017-11-27
**Interesting work at the intersection of reinforcement learning and variational inference**

**Rating:** 7
**Confidence:** 4

**Review:**

The submission tackles an important problem of learning and transferring multiple motor skills. The approach relies on using an embedding space defined by latent variables and entropy-regularized policy gradient / variational inference formulation that encourages diversity and identifiability in latent space.

The exposition is clear and the method is well-motivated. I see no issues with the mathematical correctness of the claims made in the paper. The experimental results are both instructive of how the algorithm operates (in the particle example), and contain impressive robotic results. I appreciated the experiments that investigated cases where true number of tasks and the parameter T differ, showing that the approach is robust to choice of T.

The submission focuses particularly on discrete tasks and learning to sequence discrete tasks (as training requires a one-hot task ID input). I would like a bit of discussion on whether parameterized skills (that have continuous space of target location, or environment parameters, for example) can be supported in the current formulation, and what would be necessary if not.

Overall, I believe this is in interesting piece of work at a fruitful intersection of reinforcement learning and variational inference, and I believe would be of interest to ICLR community.

---

> ### Author Response · Authors · 2018-01-05
> **Response to Reviewer 2**
>
> We thank the reviewer for their comments and suggestions.
>
> Our method does indeed support parameterized skills as suggested by the reviewer. For instance, the low-level policy could receive an embedding conditioned on a continuous target location instead of the task ID (given a suitable embedding space). It is also not limited to the multi-task setting, i.e., the number of tasks T used for training can be set to 1 (as explored in the point-mass experiments). We will add this to the discussion to the paper.

---

### Official Review · AnonReviewer1 · 2017-12-01
**This is an interesting deep reinforcement learning paper that introduces a new principled framework for learning versatile skills. This is a good paper.**

**Rating:** 7
**Confidence:** 5

**Review:**

The paper presents a new approach for hierarchical reinforcement learning which aims at learning a versatile set of skills. The paper uses a variational bound for entropy regularized RL to learn a versatile latent space which represents the skill to execute. The variational bound is used to diversify the learned skills as well as to make the skills identifyable from their state trajectories. The algorithm is tested on a simple point mass task and on simulated robot manipulation tasks.

This is a very intersting paper which is also very well written. I like the presented approach of learning the skill embeddings using the variational lower bound. It represents one of the most principled approches for hierarchical RL.

Pros:
- Interesting new approach for hiearchical reinforcement learning that focuses on skill versatility
- The variational lower bound is one of the most principled formulations for hierarchical RL that I have seen so far
- The results are convincing

Cons:
- More comparisons against other DRL algorithms such as TRPO and PPO would be useful

Summary: This is an interesting deep reinforcement learning paper that introduces a new principled framework for learning versatile skills. This is a good paper.

More comments:
- There are several papers that focus on learning versatile skills in the context of movement primitive libraries, see [1],[2],[3]. These papers should be discussed.

[1] Daniel, C.; Neumann, G.; Kroemer, O.; Peters, J. (2016). Hierarchical Relative Entropy Policy Search, Journal of Machine Learning Research (JMLR),
[2] End, F.; Akrour, R.; Peters, J.; Neumann, G. (2017). Layered Direct Policy Search for Learning Hierarchical Skills, Proceedings of the International Conference on Robotics and Automation (ICRA).
[3] Gabriel, A.; Akrour, R.; Peters, J.; Neumann, G. (2017). Empowered Skills, Proceedings of the International Conference on Robotics and Automation (ICRA).

---

> ### Author Response · Authors · 2018-01-05
> **Response to Reviewer 1**
>
> We very much appreciate the reviewer’s comments and suggestions.
>
> Regarding the comparison to other on-policy methods such as TRPO or PPO, we would like to emphasize that the presented approach is mostly independent of the underlying RL learning algorithm. In fact, it will be easier to implement our approach in the on-policy setup. The off-policy setup with experience replay that we are considering requires additional care due to the embedding variable which we also maintain in the replay buffer. In Section 5, we present all the modifications necessary to running our method in the more data-efficient off-policy setup, which we believe is crucial to running it on the real robots in the future.
>
> We would also like to thank the reviewer for pointing out the additional references - we will be very happy to include them. While some of the high-level ideas are related, there are differences both in the formulation and the algorithmic framework. An important aspect of our work is that we show how to apply entropy-regularized RL with latent variables when working with neural networks and in an off-policy setting, avoiding both the burden of using a limited number of hand-crafted features and allowing for data-efficient learning.

---

### Author Response · Authors · 2018-01-17
**Manuscript updated**

Dear reviewers,
We would like to let you know that we have updated the manuscript with the changes requested in your reviews. Thank you again for your feedback.

---

### Decision · Program_Chairs · 2018-01-29
**ICLR 2018 Conference Acceptance Decision**

**Decision:**

Accept (Poster)

**Comment:**

This is a paper introducing a hierarchical RL method which incorporates the learning of a latent space, which enables the sharing of learned skills.

The reviewers unanimously rate this as a good paper. They suggest that it can be further improved by demonstrating the effectiveness through more experiments, especially since this is a rather generic framework. To some extent, the authors have addressed this concern in the rebuttal.